# Does experiencing a traumatic life event increase the risk of intimate partner violence for young women? A cross-sectional analysis and structural equation model of data from the *Stepping Stones and Creating Futures* intervention in South Africa

Jenevieve Mannell ,[1] Nicole Minckas,[1] Rochelle Burgess,[1] Esnat D Chirwa,[2] Rachel Jewkes,[2] Andrew Gibbs [2]

¹Institute for Global Health, UCL, London, UK
²Gender and Health Division, South African Medical Research Council, Pretoria, South Africa

**Correspondence to**
Dr Jenevieve Mannell;
j.mannell@ucl.ac.uk

## ABSTRACT

**Objectives** To investigate associations and potential pathways between women's lifetime exposure to traumatic events and their recent experiences of intimate partner violence (IPV).

**Setting** South African informal settlements near Durban.
**Participants** 677 women, living in informal settlements, aged 18–30 years, currently out of school or formal employment.
**Primary and secondary outcome measures** Self-reported experiences of IPV in the past 12 months and exposure to traumatic neighbourhood events (including witnessing murder, being robbed or kidnapped, witnessing and experiencing rape).
**Results** Exposure to traumatic events was common among the 677 women surveyed. Over 70% had experienced at least one in their lifetime; one quarter (24%) had experienced 3 or more different events. Women exposed to any traumatic event had a 43% increase in the odds of experiencing IPV in comparison to those with no exposure (aOR 1.43, p≤0.000). Exposure to non-partner rape is more strongly associated with IPV than any other traumatic experience. Pathways from exposure to traumatic events and non-partner rape to recent IPV experience are mediated by a latent variable of poor mental health. Food insecurity is associated with all forms of traumatic experience, and is also indirectly associated with IPV through views by women that are unsupportive of gender equality.
**Conclusions** Women living in South African informal settlements who witness or experience traumatic events were likely to experience IPV, and this increases when women were exposed to multiple types of events. Our model suggests that experiencing traumatic events, and non-partner rape in particular, has negative effects on women's mental health in ways that may increase their vulnerability to IPV. IPV prevention interventions should consider the broader impacts of women's exposure to

---

**Strengths and limitations of this study**

► This study was the first, to our knowledge, to assess associations and potential pathways between young women's experiences of traumatic events (eg, being robbed at gun point, witnessing murder) in neighbourhoods with a high incidence of crime and the odds that they will experience recent intimate partner violence.

► Structural equation modelling provides insight into how mental health indicators mediate the relationships between young women's exposure to traumatic events and intimate partner violence, which would not have been possible using logistic regression alone.

► Study participants were randomly sampled as part of baseline data collection for a cluster randomised controlled trial and is therefore representative of the young women living in South Africa's informal settlements.

► The study relies on self-reported measures of symptoms of post-traumatic stress disorder (PTSD) and depression rather than a formal diagnosis; the results therefore reflect women's understanding of symptoms of stress and anxiety and not PTSD or depression as defined in the *Diagnostic and Statistical Manual of Mental Disorders*.

► From the questions women were asked about traumatic events in their lives, we were only able to assess whether women had experienced multiple types of events and not how many times each event may have been experienced by an individual.

neighbourhood violence and severe poverty on IPV risk in settings where these are endemic.

**Trial registration number** NCT03022370; post-results.

## INTRODUCTION

Experiencing or witnessing violence can have long-term traumatising effects that undermine women's mental health and well-being.[1–3] Experiencing multiple types of violence, such as child abuse, intimate partner violence (IPV) and/or armed conflict, has been shown to compound these negative health effects.[4–8] Our aim in this article is to contribute a better understanding of the cumulative effects of these different forms of violence and how these play out in the complex lives of women living in economically constrained settings.[9 10] This responds to calls to explore the intersectional forms of violence at work in women's lives,[11] including the simultaneous impacts of a range of structural violence(s), such as poverty, racism xenophobia and exclusion,[12] and their negative mental health consequences.[13]

Many women live in dangerous environments where the potential of experiencing community-based violence is extremely high, and studies suggest that this is associated with IPV experience. For instance, women living in settings either experiencing or that have recently experienced armed conflict are at a significantly higher risk of also experiencing IPV.[7 14 15] Investigating IPV in Liberia, Kelly and colleagues find that women living in districts experiencing conflict were almost three times as likely to experience IPV compared with individuals in districts without conflict, even 4 years after peace was declared.[16] Studies of women's experiences during armed conflict point to potential pathways through which an experience of war and conflict increases IPV. This includes poor mental health, which increases women's vulnerability to IPV as well as magnifying the negative outcomes of violence,[17] and the role of stress and other symptoms of post-traumatic stress disorder (PTSD) in increasing men's use of violence in the household.[18–21] These risk factors for IPV are often linked theoretically to the gender inequalities that drive violence against women and girls, which are magnified by the highly stressful experiences of people affected by armed conflict.[22 23]

Far less in known about the impact of violence in settings which are not conflict or post-conflict settings, but where exposure to traumatic events—such as being robbed at knifepoint, witnessing a murder or experiencing rape—are incredibly common, and the association of these experiences to IPV. Informal settlements in South Africa have a far higher incidence of traumatic events than the rest of the country, and than many other countries around the world.[24 25] In a study of men's exposure to traumatic life events in informal settlements in South Africa, witnessing or experiencing violent events was common with nearly a quarter of men having witnessed the murder of a family or friend.[26] In these same informal settlements, two-thirds (65.2%) of women are estimated to have experienced IPV.[27] The frequency with which people experience violence in informal settlements contributes to its normalisation, which is rooted in longstanding economic, racial and gender inequalities in South African society.[28] This normalisation of violence in turn impacts mental health

by upholding toxic masculinities that position strong men as desirable, and delegitimising the stress and fear women feel when experiencing acts of violence.[29 30]

To further understand intersections between different forms of violence in women's lives, we investigated the association between women's lifetime exposure to different types of traumatic events and their recent experiences of IPV (past 12 months). We set out to test three main hypotheses: (1) women who experience more types of traumatic events have a higher risk of experiencing IPV, (2) there is a dose relationship between exposure to traumatic events and IPV (eg, women who experience more types of traumatic events are more likely to experience IPV), (3) the relationship between exposure to traumatic events and IPV experiences is mediated by poor mental health.

## METHODS

### Design

Our analysis draws on women's data collected as part of the baseline for Stepping Stones and Creating Futures—a cluster randomised controlled trial of an intervention to reduce IPV and HIV vulnerability among young people living in urban informal settlements in South Africa.[31 32] Baseline data were collected between September 2015 and September 2016 in urban informal settlements near Durban.[33]

Women included in the Stepping Stones and Creating Futures (SS/CF) trial were 18–30 years old, out of school and not currently in formal employment. They were also residents of the informal settlement, although the length of time they had lived in the settlement varied. All participants were recruited by Project Empower, a local NGO with over 20 years of experience working on participatory programming in South Africa's informal settlements. Women were recruited by Project Empower through staff walking around the community and talking to potential participants, as well as referral from other participants. The project was explained in detail to participants by trained data collectors, who were not Project Empower staff, and everyone was given time to ask questions about the research before giving their written consent to participate in the survey.

### Data collection

The trial included 34 clusters with 19–21 women per cluster. All trial participants completed baseline questionnaires on a mobile phone in English, isiZulu or Xhosa, with the support of same-sex fieldworkers to assist in cases of low literacy. Questionnaires were tailored for women (a separate questionnaire was provided for men) and had logical skips patterns to facilitate survey completion. Questionnaires took between 45 min and 1.5 hours to complete. Participants were reimbursed for completing the questionnaire: R100 (~US$7) for those in the intervention arm and R300 (~US$21) in the control arm,

because of the differential benefit of being involved in the intervention.

## Patient and public involvement

The local partner organisation, Project Empower, participated in framing the research questions and planning the SS/CF intervention. They were active collaborators in the study, including quantitative and qualitative data collection and formulating recommendations from the analysis. Project Empower engaged community networks to ensure permission for community access and provided feedback to community leaders and organisations on the results of the study. The results of the trial have been disseminated to the communities involved. Our analysis will also be shared with young people living in informal settlements as part of a broader commitment to their active participation in interpreting findings about the health issues that affect their lives.[34 35]

## Measures

IPV was assessed using eight behaviourally specific questions about women's experiences, five about physical IPV and three about sexual IPV, with response options for each item: 'never', 'once' 'few' or 'many times'. These questions were based on the WHO multicountry study survey on violence against women,[36] which has been adapted for South Africa.[37] A positive response to any of the questions was coded as having experienced IPV in the past 12 months.

Traumatic experiences, as the exposure variable, was compiled from twelve questions about women's past experiences of traumatic events in their lifetime. This included six questions about witnessing the murder of a friend/ family member, witnessing the murder of a stranger, witnessing an armed attack, being robbed at knife or gun point, being kidnapped, and witnessing someone being raped, and six questions about experiences of non-partner rape (forced sex, attempted forced sex, forced sex while under the influence of alcohol or drugs, forced sex by multiple men at the same time, forced sex by multiple men at the same time while under the influence of alcohol or drugs, and forced sex after agreeing to have sex with someone else). Possible responses to each item were yes or no. We generated a binary variable for any non-partner rape that combined the six related items. We then summarised women's tramautic experiences in three ways. First, we generated a binary variable for never/any experience of trauma. Second, we generated a variable classifying women into those who had (1) never experienced a traumatic event, (2) experienced 1–2 types of events, and (3) experienced 3 or more types of traumatic events. Third, we generated a variable that summed all the types of trauma into a score (range 0–7).

Women's mental health was assessed via measures of depressive symptoms, symptoms of PTSD, and alcohol use. For depressive symptoms, the Centre for Epidemiological Studies Depression Scale (CES-D) was used, which asks 20 questions about depressive symptoms in the past week with responses on a four-point Likert scale ranging from 'never' to 'most or all of the time'.[38] Values associated with the Likert scale were summed to create a CES-D score (range 0–57, α=0.88). Symptoms of PTSD were assessed using the Harvard Trauma Questionnaire,[39] which includes 16 questions about feelings in the past week related to traumatic events in the past with responses on a four-point Likert scale ranging from 'not at all' to 'extremely.' Values were summed to create a PTSD score (range 0–45, α=0.9), with a clinical cut-off score of 2.5. Harmful alcohol use was assessed using the Alcohol Use Disorder Identification Test (AUDIT) scale, which asks 10 questions about past year alcohol use. Values associated with the Likert scale were summed to create an AUDIT score (range 0–38, α=0.81). Harmful alcohol use was defined as an AUDIT score of eight or more.[40]

Other measures included demographic data on age, education level (primary/ secondary), and relationship status (married/ living with a partner, boyfriend, no relationship). Food insecurity was assessed using the Household Hunger Scale,[41] which asks how often in the past month there was no food to eat in their household, how often a member of the household had gone to sleep hungry, and how often a member of the household had not eaten for a whole day and night because of no food. Possible responses were on a four-point Likert scale (never, sometimes, often, very often), and summed values were recoded as three categorical variables: little or no hunger/moderate hunger/severe hunger. Gender views were assessed using the Gender Equitable Men's scale adapted for South Africa. This included 20 statements about the role and expectations of men and women with a four-point Likert scale from 'Strongly disagree' to 'Strongly agree.' Higher scores correspond to views that are less supportive of gender equality (range 2–54, α=0.86).

## Analysis

All analyses were performed in STATA V.16.1 and considered clustering of the survey data using the SVY command. We first provide descriptive statistics for the sample providing numbers and percentages. Second, we calculated the prevalence of the eight traumatic events asked about in the survey, and the various sums for overall trauma experiences.

We then carried out descriptive analyses of the relationship between socioeconomic and mental health data and exposure to traumatic events. We provide percentages and 95% CIs, or mean and 95% CI as appropriate and tested for significant differences using t-tests for continuous variables and $\chi^2$ tests for categorical variables and report p values.

We used logistic regression to estimate unadjusted and adjusted associations between exposure to traumatic events and IPV experiences in the past 12 months. We first assessed whether each individual traumatic event was associated with IPV, then whether never/any, then never, 1, 2, 3+. We reported ORs, 95% CI and p values.

Adjusted models included the covariates age, education, relationship status, harmful alcohol use, food insecurity and gender views. Due to ORs overinflating the association between variables with non-rare events, we re-ran the analysis as a supplementary analysis, using Poisson regression to calculate relative risk ratios, using the same modelling approach as for logistic regression.

Second, we built a structural equation model (SEM) to assess whether there were direct and mediated pathways between lifetime traumatic events and past 12 month IPV experience. To build the SEM, we first identified potential pathways drawing on the theoretical literature available. Given the qualitative difference between being raped oneself and witnessing a traumatic event (such as murder or rape), we constructed a latent variable for non-partner rape and separated this from other experiences of traumatic events. We then regressed each variable included in the draft model to all other variables they were hypothesised to be associated with (ie, continuous outcome scores for food insecurity, gender views, depression, PTSD, and harmful alcohol use), and included only those pathways that were significant (p<0.05). Based on a previous analysis of the same dataset,[27] we constructed a latent variable for food insecurity by separating questions from the household hunger scale into three distinct variables as part of the model. We then assessed how well our model predicted the sample variance-covariance matrix by examining indices of model fit including the standardised root mean squared residual (<0.1), root mean squared error of approximation (RMSEA <0.05), comparative fit index (CFI >0.95), and Tucker-Lewis Index (TLI>0.95). We modified the model using the MINDICES (modification indices) command in STATA to identify correlations between error terms with a good trade-off between improved model fit and an increase in df (eg, modification indices>3.84). We then covaried error terms where we could justify this theoretically.

## RESULTS

At baseline, 677 young women were recruited and provided data. Roughly one-third of the women (30.4%) had completed secondary education (table 1). Two-thirds were in a relationship: 63.5% had a boyfriend but were not living with them, and 18.2% were either living with a partner or married. Food insecurity was also common with 50.1% of women living in households experiencing moderate food insecurity, and an additional 31.3% with severe food insecurity. Approximately one-fifth of women experienced poor mental health: 21% of women were above the mean of 2.5 on the PTSD scale; 45.2% met the cut-off measure of 20/21 for depression; and 23% of women met the cut-off measure of 8 or more for harmful alcohol use.

Lifetime experience of traumatic events was common in the lives of these women (table 2). Over a third had been robbed at knife or gun point (35.2%), and roughly the same number had witnessed an armed attack (34.6%). A quarter (26.1%) had witnessed the murder of a stranger or someone they know, while almost a fifth (17.6%) had witnessed the murder of a friend or family member. One in ten (10.5%) had witnessed someone being raped, and nearly a third (30.4%) had been raped themselves. Over 70% of women had experienced at least one of these traumatic events, while 24% had experienced three or more different types.

While there is a significant difference in the number of different traumatic events experienced by women who have completed secondary education in comparison to those who have not, there is not a significant difference according to age, relationship status or food insecurity. Gender views are also not significantly associated with an increase in exposure to different traumatic events. However, the variables signifying common mental disorders (depression, symptoms of PTSD, and harmful alcohol use) are all significantly associated with an increase in the number of different traumatic experiences.

Table 3 reports results from the logistic regression. Non-partner rape is the only traumatic experience that has a significant association with IPV experience in the past year when controlling for age, education, relationship status, food insecurity, harmful alcohol use and gender views. Women exposed to any traumatic event other than experiencing non-partner rape have a 43% increase in the odds of experiencing IPV in the past year in the adjusted analysis, compared with those with no trauma exposure (aOR 1.43). When non-partner rape is included as a traumatic experience in the model, women have an 83% increase in the odds of experience IPV in the past year, compared with those with no trauma exposure (aOR 1.83). There also appears to be a relationship whereby the odds of experiencing IPV increases as women are exposed to one or two types of traumatic events in comparison to those who did not experience any such events (1 event, aOR 1.66; 2 events, aOR 2.47). However, this dose relationship between exposure to different traumatic events and IPV experience tapers off when more than two types of traumatic events have been experienced (3+events, aOR 1.71). The Poisson regression shows the same associations in unadjusted and adjusted models, with the relative risk ratios being lower in magnitude than the logistic regression.

Table 4 presents the direct associations between all variables in preparation for the SEM with significant associations (p<0.05) highlighted in bold. Nearly all associations were significant with the exception of the association between food insecurity and alcohol abuse, and several associations with gender attitudes (including exposure to traumatic events, alcohol abuse, depression and PTSD). These associations were removed from our SEM.

In the SEM, the importance of non-partner rape as a traumatic experience meant that it needed to be separated as a variable from other types of traumatic events to attain a good model fit. As shown in figure 1, the association between exposure to traumatic events (including the covaried effects of non-partner rape) and IPV in the past

**Table 1** Women's sociodemographic and baseline characteristics, by exposure to traumatic events

| | N | % | No traumatic experience (n=185) Mean/% (95% CI) | 1–2 traumatic experiences (n=326) Mean/% (95% CI) | 3+ traumatic experiences (n=166) Mean/% (95% CI) | P value |
|---|---|---|---|---|---|---|
| **Age** | | | | | | 0.97 |
| 18–20 | 142 | 21.0% | 20.0 (14.8 to 26.4) | 22.09 (17.9 to 26.9) | 19.9 (14.5 to 26.7) | |
| 21–25 | 313 | 46.2% | 47.6 (40.4 to 54.8) | 45.4 (40.0 to 50.9) | 46.4 (38.9 to 54.1) | |
| 26–30 | 222 | 32.8% | 32.4 (26.0 to 39.6) | 32.5 (27.6 to 37.8) | 33.73 (26.9 to 41.3) | |
| **Education** | | | | | | 0.02 |
| Secondary complete | 206 | 30.4% | 62.7 (55.7 to 69.3) | 69.9 (64.8 to 74.6) | 76.5 (69.5 to 82.3) | |
| Secondary incomplete/no secondary | 471 | 69.6% | 37.3 (30.8 to 44.3) | 30.1 (25.4 to 35.2) | 23.5 (17.7 to 30.5) | |
| **Relationship status** | | | | | | 0.82 |
| Married/cohabitating | 123 | 18.2% | 18.9 (13.9 to 25.2) | 17.8 (14.0 to 22.3) | 18.1 (12.9 to 24.7) | |
| Boyfriend | 430 | 63.5% | 62.7 (55.5 to 69.4) | 65.3 (60.0 to 70.3) | 60.8 (53.1 to 68.0) | |
| No relationship | 124 | 18.3% | 18.4 (13.5 to 24.5) | 16.87 (13.2 to 21.3) | 21.1 (15.5 to 28.0) | |
| **Food insecurity** | | | | | | 0.16 |
| Mild (no/little hunger) | 126 | 18.6% | 22.7 (17.3 to 29.2) | 17.8 (14.0 to 22.3) | 15.7 (10.9 to 22.0) | |
| Moderate | 339 | 50.1% | 51.9 (44.6 to 59.1) | 50.3 (44.9 to 55.7) | 47.6 (40.1 to 55.2) | |
| Severe | 212 | 31.3% | 25.4 (19.6 to 32.2) | 31.9 (27.0 to 37.2) | 36.8 (29.8 to 44.3) | |
| **Common mental disorders** | | | | | | |
| Depression (score)* | 677 | NA | 17.4 (16.0 to 18.8) | 22.3 (21.1 to 23.4) | 23.1 (21.4 to 24.8) | |
| PTSD (score)* | 677 | NA | 1.6 (1.5 to 1.7) | 1.9 (1.9 to 2.0) | 2.1 (2.0 to 2.2) | |
| Alcohol abuse (score)* | 677 | NA | 0.1 (0.1 to 0.2) | 0.2 (0.2 to 0.3) | 0.3 (0.3 to 0.4) | |
| **Gender views** | | | | | | |
| (score) | 677 | NA | 25.2 (23.8 to 26.5) | 26.0 (25.0 to 26.0) | 25.4 (23.9 to 27.0) | |

*P value<0.001: levels of significance for continuous variables calculated using t-tests (no trauma experience as comparison).
NA, Not applicable; PTSD, post-traumatic stress disorder.

**Table 2** Proportion of women who experienced traumatic events by type of event and frequency

| | N | % (95% CI) |
|---|---|---|
| **By type of traumatic event** | | |
| Witnessed murder friend | 119 | 17.6 (14.7 to 20.5) |
| Witnessed murder stranger | 117 | 26.1 (22.8 to 29.5) |
| Witnessed armed attack | 234 | 34.6 (31.0 to 38.2) |
| Robbed at gun/ knife point | 238 | 35.2 (31.5 to 38.8) |
| Kidnapped | 32 | 4.7 (3.1 to 6.3) |
| Witnessed rape | 71 | 10.5 (8.2 to 12.8) |
| Experienced non-partner rape | 205 | 30.3 (26.8 to 33.8) |
| **Multiple traumatic events** | | |
| Yes | 492 | 72.7 (69.3 to 76.0) |
| **Number of different traumatic events** | | |
| None | 185 | 27.3 (24.1 to 30.8) |
| 1 traumatic event | 209 | 30.87 (27.5 to 34.5) |
| 2 traumatic events | 117 | 17.28 (14.6 to 20.3) |
| 3 or more traumatic events | 166 | 24.5 (21.4 to 27.9) |

12 months was mediated by indicators of poor mental health, specifically symptoms of depression, PTSD and alcohol abuse, which were all increased by trauma experience and which were directly associated with increased IPV. Increased symptoms of poor mental health were also mediated by increased food insecurity. Food insecurity

was also associated with IPV experience mediated by inequitable gender views. The SEM was a good fit with the data (RMSEA 0.046; CFI 0.97; TLI 0.961). All standardised coefficients for the SEM are presented in table 5.

## DISCUSSION

These results support our hypothesis that women living in informal settlements who experience traumatic events are at a higher risk of experiencing IPV. There was also a large effect size for those experiencing two or more different traumatic events, in comparison to only one event. This acknowledges how women's exposure to multiple and overlapping violence(s) (including violence outside of the household) impacts their risk of IPV and poor mental health outcomes for women, building on the already well-recognised consideration of the role of men's exposure to violence in their perpetration of IPV.[14 42]

Both the logistic regression and the SEM highlight the overwhelming importance of non-partner rape as a traumatic event in women's risk of experiencing IPV. Non-partner rape is more strongly associated with IPV experience than any other traumatic event assessed in our analysis. This highlights the importance of non-partner rape in women's risk of experiencing IPV, confirming findings from other studies that have highlighted associations between non-partner rape and an increased risk of IPV.[43] This may be explained by the devastating consequences of non-partner rape for women's mental health and the importance of self-blame[44] and disengagement

**Table 3** Associations between exposure to traumatic events and intimate partner violence in the past 12 months

| | Logistic | | Poisson | |
|---|---|---|---|---|
| Variable | OR (95% CI) | aOR (95% CI) | RRR (95% CI) | aRRR (95% CI) |
| **Individual traumatic event** | | | | |
| Witnessed murder friend | 1.23 (0.80 to 1.88) | 1.00 (0.63 to 1.59) | 1.07 (0.93 to 1.23) | 1.02 (0.89 to 1.16) |
| Witnessed murder stranger | 1.02 (0.71 to 1.47) | 0.92 (0.62 to 1.37) | 1.01 (0.89 to 1.14) | 0.98 (0.87 to 1.11) |
| Witnessed armed attack | **1.40 (1.00 to 1.97)\*** | 1.34 (0.93 to 1.92) | **1.12 (1.00 to 1.25)\*** | 1.10 (0.99 to 1.23) |
| Robbed at gun/ knife point | 1.37 (0.98 to 1.92) | 1.19 (0.82 to 1.76) | 1.11 (1.00 to 1.24) | 1.06 (0.95 to 1.19) |
| Kidnapped | 1.64 (0.72 to 3.73) | 0.85 (0.36 to 1.93) | 1.16 (0.94 to 1.43) | 0.97 (0.79 to 1.18) |
| Witnessed rape | **1.96 (1.09 to 3.51)\*** | 1.58 (0.85 to 2.92) | **1.22 (1.06 to 1.40)\*\*** | 1.13 (0.98 to 1.29) |
| Experienced non-partner rape | **2.82 (1.91 to 4.16)\*\*\*** | **2.35 (1.54 to 3.57)\*\*\*** | **1.36 (1.23 to 1.51)\*\*\*** | **1.27 (1.14 to 1.41)\*\*\*** |
| **Multiple traumatic events** | | | | |
| Any trauma (excluding non-partner rape) | **1.61 (1.16 to 2.23)\*\*** | **1.43 (1.00 to 2.04)\*** | **1.19 (1.05 to 1.35)\*\*** | **1.14 (1.01 to 1.28)\*** |
| Any trauma (including non-partner rape) | **2.11 (1.49 to 2.98)\*\*\*** | **1.83 (1.26 to 2.66)\*\*** | **1.33 (1.15 to 1.55)\*\*\*** | **1.25 (1.08 to 1.45)\*\*** |
| **Number of different traumatic events** | | | | |
| 1 traumatic event | **1.76 (1.17 to 2.65)\*\*\*** | **1.66 (1.08 to 2.57)\*** | **1.26 (1.06 to 1.49)\*\*** | **1.22 (1.03 to 1.43)\*** |
| 2 types of traumatic events | **2.88 (1.74 to 4.78)\*\*\*** | **2.47 (1.43 to 4.27)\*\*** | **1.45 (1.23 to 1.72)\*\*\*** | **1.35 (1.14 to 1.60)\*\*** |
| 3+ types of traumatic events | **2.17 (1.39 to 3.36)\*\*\*** | **1.71 (1.05 to 2.78)\*** | **1.34 (1.14 to 1.59)\*\*\*** | **1.24 (1.05 to 1.46)\*** |

Model adjusted for age, education, relationship status, food insecurity, harmful alcohol use and gender views.
Boldface indicates statistical significance (\*p<0.05; \*\*p<0.01; \*\*\*p<0.001).

**Table 4** Assessing potential pathways of association

| | Traumatic events | IPV past 12 months | Non-partner rape | Food insecurity | Alcohol abuse | Depress | PTSD |
|---|---|---|---|---|---|---|---|
| IPV past 12 months | **0.052** | | | | | | |
| Non-partner rape | **0.687** | **0.028** | | | | | |
| Food insecurity | **0.052** | **0.091** | **0.020** | | | | |
| Alcohol abuse | **0.040** | **0.356** | **0.021** | 0.015 | | | |
| Depression | **0.017** | **0.185** | **0.010** | **0.053** | **0.163** | | |
| PTSD | **0.035** | **0.211** | **0.011** | **0.454** | **0.203** | **0.779** | |
| Gender views | 0.003 | **0.081** | 0.002 | **0.068** | 0.019 | 0.023 | 0.039 |

Associations significant at 95% CI in bold.
IPV, intimate partner violence; PTSD, post-traumatic stress disorder.

coping strategies (ie, passive reactions and avoidance[45] in increasing their vulnerability to other forms of violence.[46] It also points to a need for further research to investigate positive mental health supports (ie, engagement coping strategies, gender awareness, etc) that could be provided to women following experiences of non-partner rape that may help in reducing their risk of IPV. For example, recent work has highlighted the ability for such interventions to reduce symptoms of poor mental health, as well as increase self-efficacy and collective agency, in response to the social-structural challenges that increase women's risk of IPV (including food insecurity).[47]

Our analysis also shows the importance of indicators of poor mental health (including depression, symptoms of PTSD, and harmful alcohol use) in mediating the relationship between experiencing traumatic events and IPV, as shown in the SEM. This is consistent with literature on the role of symptoms of PTSD in increasing women's risk of physical and sexual revictimisation,[46 48] and confirms findings from other studies of the importance of poor mental health and substance abuse as risk factors for IPV.[49 50] While our analysis has focused on poor mental health resulting from exposure to interpersonal neighbourhood-based acts of violence, it supports findings from other studies that have examined the increased IPV risk of exposure to traumatic events such as natural disasters.[51 52] The evidence on interpersonal violence and disasters are similar in suggesting that the relationship between poor mental health and IPV is bidirectional, whereby symptoms of poor mental health leads to IPV, which in turn worsens women's mental health.[53 54]

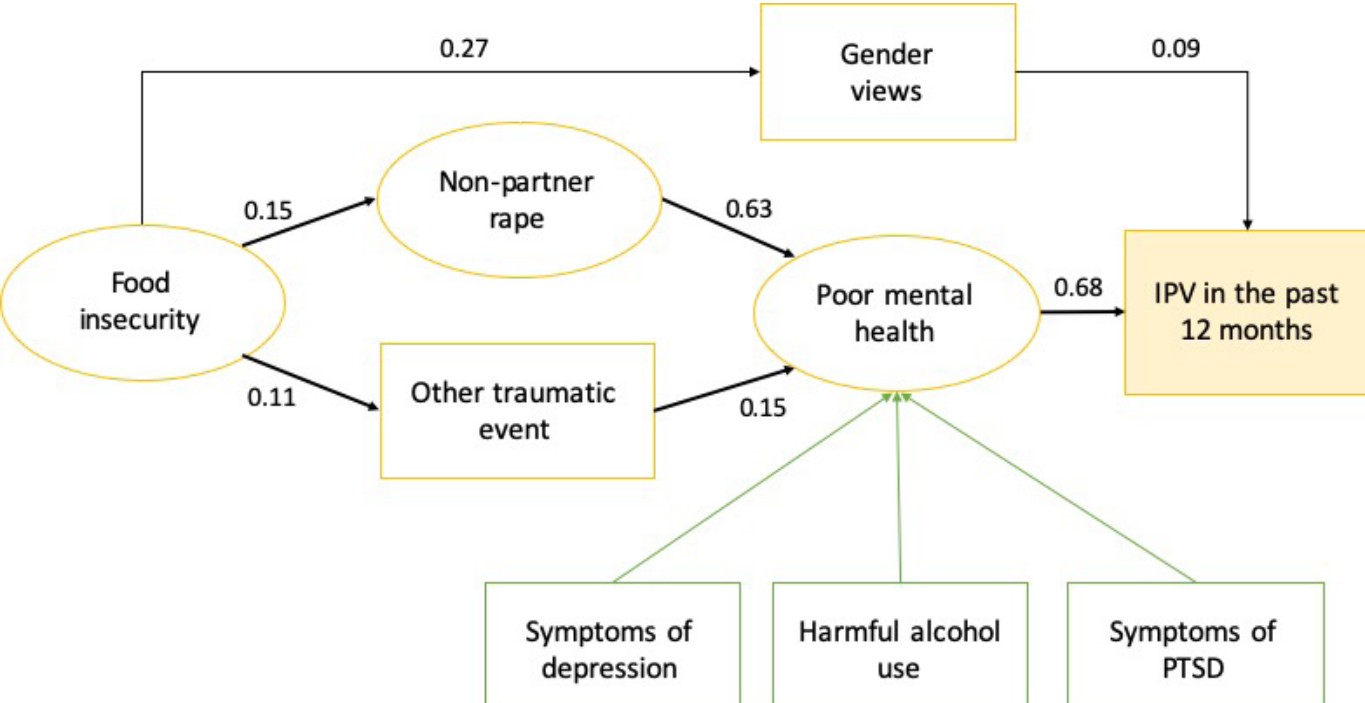

**Figure 1** Structural equation model of pathways between exposure to traumatic events and IPV in the past 12 months (standardised coefficients).

**Table 5** Structural equation model

| Parameter<br>Direct effects | Standardised<br>coefficients | SE | z | P>\|z\| | 95% CI | |
|---|---|---|---|---|---|---|
| Food insecurity→trauma | 0.11 | 0.04 | 2.70 | 0.007 | 0.03 | 0.19 |
| Food insecurity→ Non-partner rape | 0.15 | 0.04 | 3.58 | 0.000 | 0.07 | 0.24 |
| Food insecurity→Gender views | 0.27 | 0.04 | 7.01 | 0.000 | 0.19 | 0.34 |
| Trauma→Poor mental health | 0.15 | 0.04 | 3.36 | 0.000 | 0.06 | 0.23 |
| Non-partner rape→poor mental health | 0.63 | 0.04 | 15.43 | 0.000 | 0.55 | 0.71 |
| Poor mental health→IPV | 0.68 | 0.03 | 20.65 | 0.000 | 0.62 | 0.75 |
| Gender views→IPV | 0.09 | 0.03 | 2.69 | 0.007 | 0.02 | 0.15 |
| **Latent variable: non-partner rape** | | | | | | |
| Forced sex | 0.65 | 0.03 | 25.28 | 0.000 | 0.59 | 0.70 |
| Attempted forced sex | 0.54 | 0.03 | 17.58 | 0.000 | 0.48 | 0.59 |
| Forced sex, alcohol or drugs | 0.75 | 0.02 | 32.42 | 0.000 | 0.71 | 0.79 |
| Forced sex, multiple men | 0.71 | 0.02 | 30.30 | 0.000 | 0.67 | 0.76 |
| Forced sex, multiple men, alcohol or drugs | 0.82 | 0.08 | 45.00 | 0.000 | 0.79 | 0.86 |
| Forced sex after agreeing with someone else | 0.69 | 0.02 | 27.92 | 0.000 | 0.64 | 0.74 |
| **Latent variable: poor mental health** | | | | | | |
| Symptoms of depression | 0.45 | 0.04 | 11.63 | 0.000 | 0.38 | 0.53 |
| Harmful alcohol use | 0.61 | 0.03 | 17.51 | 0.000 | 0.54 | 0.67 |
| Symptoms of PTSD | 0.49 | 0.04 | 13.27 | 0.000 | 0.43 | 0.57 |
| **Latent variable: food insecurity** | | | | | | |
| No food to eat | 0.78 | 0.02 | 32.70 | 0.000 | 0.68 | 0.77 |
| Going to sleep hungry | 0.93 | 0.02 | 54.15 | 0.000 | 0.89 | 0.96 |
| 24 hours without eating | 0.75 | 0.02 | 35.14 | 0.000 | 0.71 | 0.79 |

Furthermore, our analysis shows how food insecurity (as a proxy for poverty) is a contributing factor in women's exposure to traumatic events. In the SEM, food insecurity plays a role in increasing both non-partner rape and other traumatic events in women's lives.

Moreover, the association between food insecurity and IPV experience is mediated by gender views unsupportive of women's equality, a pathway corroborated by evidence that food insecurity may be an indicator of women's economic dependence on men with implications for their ability to leave violent relationships.[27] The role of food insecurity as a compounding factor in women's exposure to trauma has important implications for thinking about poverty as an additional form of structural violence in women's lives,[55] and how this in turn compounds the risk of IPV in women's relationships. This contributes important insights to current understandings of the effects of multiple experiences of trauma or *polyvictimi-sation*[56–58] by highlighting the need to explore violence happening at a structural as well as an interpersonal level.

### Limitations
This study has some limitations. Our data rely on self-reported measures of traumatic events, mental health and IPV. In the case of IPV and traumatic events, questions have been framed around specific acts of violence (hitting, slapping, kicking, etc) consistent with recognised best practice in the field of violence against women. However, this resulted in a limited number of questions being asked about the types of traumatic events women had experienced and likely did not reflect the full range of events that they potentially experience in their lives. The way in which questions about trauma were asked (eg, *have you experienced any of this list of possible events?*) means that we were only able to assess whether women had experienced multiple types of events rather than how many times each event may have been experienced by an individual. Moreover, the data do not tell us if the events women experienced were actually traumatising or if they led to a mental health disorder. The survey did not ask about diagnosis and self-reporting mental health measures are limited in their diagnostic potential. The study was also cross-sectional and women were not asked about when the traumatic event occurred in their life. It is therefore possible (although unlikely) that women's experiences of traumatic events happened more recently than their experience of IPV in the past year.

### CONCLUSIONS
Our results highlight the need for interventions that comprehensively address multiple forms of violence, with particular attention to non-partner rape, as a means of

reducing this traumatic event in and of itself, and as a means of reducing IPV. In settings with extremely high rates of violence, such as South Africa's informal settlements, women are often exposed to a range of traumatic events, and this has negative consequences for their mental health in ways that put them at increased risk of IPV. As a consequence, IPV interventions in these settings require comprehensive mental health strategies and supports as an integral part of intervention activities. Working through women's trauma is not only an optional add-on to IPV prevention interventions, but an integral resource for reducing women's risk of IPV. Attention to men's trauma is also important for IPV reduction interventions, particularly given the evidence that the men who commit non-partner rape are often the same as those who perpetuate IPV, and that this is often associated with their own experiences of child abuse and neglect.[59] However, it is important to remember that attention to poor mental health as an IPV prevention strategy also needs to be considered alongside the urgent need for structural interventions that address the contexts of violence and poverty as a key driver of a multiple intersecting forms of violence in this context.

**Contributors** JM and AG conceptualised the paper. The analysis was completed by JM with assistance from EDC and AG, who verified the results at multiple time points. AG, RJ and EDC were involved in the cluster RCT that provided the baseline data for this analysis. There was substantial intellectual input from RJ, NM and RB during multiple rounds of analysis and drafts of the article. RB also contributed specifically to the interpretation of the analysis of the mental health results. JM is the guarantor of the work outlined in this paper.

**Funding** The initial study was funded through the What Works To Prevent Violence? A Global Programme on Violence Against Women and Girls (VAWG) funded by the UK Government's Department for International Development (DFID), and managed by the South African Medical Research Council. However, the views expressed do not necessarily reflect the department's official policies and the funders had no role in study design; collection, management, analysis, and interpretation of data; writing of the report; and the decision to submit the paper for publication. The secondary data analysis performed for this article was supported by UKRI MRC Global Challenge Research Fund MR/T029803/1 and UKRI Future Leaders Fellowship MR/S033629/1.

**Competing interests** None declared.

**Patient and public involvement** Patients and/or the public were involved in the design, or conduct, or reporting, or dissemination plans of this research. Refer to the Methods section for further details.

**Patient consent for publication** Consent obtained directly from patient(s)

**Ethics approval** Ethical approval was obtained from the University of Kwa-Zulu Natal (BFC043/15) and the South African Medical Research Council (EC006-2/2015). All participants gave their informed consent to participate in the survey before taking part.

**Provenance and peer review** Not commissioned; externally peer reviewed.

**Data availability statement** Data are available in a public, open access repository. Deidentified data set for this project is available from https://medat.samrc.ac.za/index.php/catalog/23 managed by the South African Medical Research Council.

**ORCID iDs**
Jenevieve Mannell http://orcid.org/0000-0002-7456-3194
Andrew Gibbs http://orcid.org/0000-0003-2812-5377

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
