## [Reviewer comments · BMJ Open]

ARTICLE DETAILS

TITLE (PROVISIONAL)	Does experiencing a traumatic life event increase the risk of intimate partner violence for young women? A cross sectional analysis and structural equation model of data from the Stepping Stones and Creating Futures intervention in South Africa
AUTHORS	Mannell, Jenevieve; Minckas, Nicole; Burgess, Rochelle; Chirwa, Esnat; Jewkes, Rachel; Gibbs, Andrew

VERSION 1 – REVIEW

REVIEWER	Watkins, Laura Emory University School of Medicine, Psychiatry and Behavioral Sciences
REVIEW RETURNED	14-Jun-2021

GENERAL COMMENTS	The manuscript “Experiencing traumatic life events increases young women’s risk of recent intimate partner violence in South African informal settlements: A cross sectional analysis and structural equation model” examined the relationships between trauma exposure and intimate partner violence in a sample of South African women. Findings indicate that a history of non-partner rape is associated with intimate partner violence. Results have implications for prevention of and intervention with interpersonal violence. However, I noted several areas of concern or in need of clarification or elaboration, which are described below. 1) I noted several areas of the SEM analyses in need of clarification. a. The authors state that they constructed a latent variable for non-partner rape. However, from prior descriptions it sounded as though this variable was binary and one item, which would be an observed variable rather than latent. Latent variables are inferred from multiple other variables that are observed. b. Please provide factor loadings for the poor mental health indicators. c. In addition, the figure indicates that food insecurity is a latent variable. If this is the case, please identify the indicators of this variable and their factor loadings. 2) The authors note that this was the first study to examine associations between traumatic events and intimate partner violence. However, there are other studies that examine past traumatic events and IPV (e.g., Iverson et al., 2013 published in Journal of Traumatic Stress), which are worth including and comparing to findings from the current study. 3) Did the authors consider examining sexual IPV and physical IPV separately? Please provide rationale for combining these two types of IPV.
---

	4) Please clarify if PTSD symptoms were linked to a specific traumatic event. Relatedly, is there a cut-off or clinical indicator for the PTSD measure? 5) I recommend stating “more types of traumatic events” rather than “more traumatic events” throughout the manuscript as the authors were not able to assess the frequency of the events. 6) The traumatic events assessed appear to all be interpersonal in nature (e.g., does not include natural disaster, fire/explosion, serious accident). This may be worth discussing and highlighting throughout the manuscript.
--	--

REVIEWER	McClair, Tracy Population Council
REVIEW RETURNED	28-Oct-2021

GENERAL COMMENTS	Overall comments This article adds an important contribution to the literature on if and how women’s experiences of traumatic life events are associated with recent intimate partner violence. The structural equation model in particular helps elucidate some of the pathways of these associations, adding much more nuance and insight into what is already known about this topic. Further, the article is very well-written and positions the current study within the literature that exists and acknowledges the limitations of the analysis as well. Specific comments  1. Can the authors add in information about how participant consent was obtained? 2. Did the authors consider using log binomial regressions to calculate relative risk ratios instead of odds ratios? I would suggest looking into this since the outcome of experiencing any traumatic event is not rare. 3. In Table 3, the authors should consider using a Bonferroni correction or another correction method to reduce the likelihood of spurious associations due to many analytical models using the same data. 4. I am wondering if the authors tried excluding “witnessed an armed attack” from the any trauma variables in Table 3. “Witnessed an armed attack” is significant at $p < 0.05$, even though the odds ratio overlaps 1.00. I think it is important to see if that’s what is driving the association between “any trauma excluding non-partner rape” with IPV, since the other individual traumatic events are not significantly associated with IPV – it’s not intuitive that “any trauma excluding non-partner rape” would be significantly associated with IPV unless essentially solely driven by witnessing an armed attack, in which case, this would have implications for framing the results and these analyses. The same applies to the 3-category variable of number of different traumatic events – could the authors conduct further analyses to determine if the majority of those experiencing 1 or 2 traumatic events are those who experienced non-partner rape and witnessed an armed attack? This may help paint a more accurate picture and allow for a more specific discussion of the results. Relatedly, if “witnessing an armed attack” is driving the association between “any trauma excluding non-partner rape” and IPV, the authors could create a 4-level variable – experienced non-partner
--

	rape (and did not witness an armed attack), witnessed an armed attack (and did not experience non-partner rape), both, and neither – to investigate these associations further. 5. I recommend the authors add to the discussion of poorer mental health as a pathway of traumatic life events to IPV, specifically, how the promotion of and support for better mental health may be linked with self-worth, resilience, empowerment, and more gender equitable views, which in turn could be linked to a reduction in the experiences of IPV. Though I would be careful to phrase this in a way that is not victim blaming, but rather supportive of women and women’s health. This could also be mentioned as an area for future research. 6. The authors may want to incorporate into the discussion (or mention that future research is needed) as to how these results link to the literature on IPV perpetration and male partners. Is it possible that the men who are perpetrating non-partner rape are also perpetrating IPV? This would have implications for interventions to reach these men to stop/reduce their perpetration of violence, whether related to their partners, non-partners, or both. 7. In the first sentence of the conclusions, I suggest adding language to claim that reducing non-partner rate is important in and of itself, rather than focusing on it as a means of reducing IPV. This sentence may come off as a bit insensitive as written. For example, the authors could write: “Our results highlight the need for interventions that comprehensively address multiple forms of violence, with particular attention to non-partner rape, as a means of reducing this traumatic event in and of itself, and as a means of reducing IPV.”
--	--

VERSION 1 – AUTHOR RESPONSE

Reviewer 1 Dr. Laura Watkins, Emory University School of Medicine	
1) I noted several areas of the SEM analyses in need of clarification. a. The authors state that they constructed a latent variable for non-partner rape. However, from prior descriptions it sounded as though this variable was binary and one item, which would be an observed variable rather than latent. Latent variables are inferred from multiple other variables that are observed.	Non-partner rape is a latent variable composed of 6 indicators of forced sex, sex under the influence of drugs or alcohol, forced sex from multiple men at the same time, and variations of these themes. We have modified the text under ‘methods’ to make this clearer in the manuscript and have also included relevant factor loadings in table 5.
b. Please provide factor loadings for the poor mental health indicators.	This has been added to table 5 in the manuscript.
c. In addition, the figure indicates that food insecurity is a latent variable. If this is the case, please identify the indicators of this variable and their factor loadings.	Food insecurity is a latent variable composed of three indicators, focused on (1) having no food to eat, (2) going to sleep hungry, and (3) going 24 hours without eating. We have included the relevant factor loadings in table 5.
2) The authors note that this was the first study to examine associations between traumatic events and intimate partner violence. However, there are other studies that examine past traumatic events and IPV	Thank you for pointing us to this very helpful paper. We have added it as a reference along with additional text to the discussion. We have also clarified in the summary bullet points that our

(e.g., Iverson et al., 2013 published in Journal of Traumatic Stress), which are worth including and comparing to findings from the current study.	study is unique in exploring traumatic events occurring in high-crime neighbourhoods as the more specific contribution of our analysis to the broader literature.
3) Did the authors consider examining sexual IPV and physical IPV separately? Please provide rationale for combining these two types of IPV.	We combined these two types of IPV because there was potential for misclassification (where a woman may experience both forms at once, but only report on one). We could not disaggregate into a combined variable (none, physical only, sexual only, both) because of the small number (38/677) reporting only experiencing sexual IPV.
4) Please clarify if PTSD symptoms were linked to a specific traumatic event. Relatedly, is there a cut-off or clinical indicator for the PTSD measure?	PTSD symptoms were assessed using the Harvard Trauma Questionnaire (HTQ) and were not linked to a specific event. We used a clinical cut-off score of 2.5 to identify PTSD, as recommended by the HTQ. We have added text to the manuscript to make this clearer.
5) I recommend stating “more types of traumatic events” rather than “more traumatic events” throughout the manuscript as the authors were not able to assess the frequency of the events.	This change has been made throughout the manuscript.
6) The traumatic events assessed appear to all be interpersonal in nature (e.g., does not include natural disaster, fire/explosion, serious accident). This may be worth discussing and highlighting throughout the manuscript.	Thanks for this comment. We have made changes throughout the manuscript to be more specific about the interpersonal nature of the violence we are discussing. We have also added the following sentence to the discussion: “While our analysis has focused on poor mental health resulting from exposure to interpersonal neighbourhood-based acts of violence, it supports findings from studies that have examined the increased IPV risk of exposure to other types of traumatic events, including natural disasters.”
Reviewer: 2 Dr. Tracy McClair, Population Council	
1. Can the authors add in information about how participant consent was obtained?	We have added more details about the informed consent procedures to the manuscript, specifically: “Women were recruited by Project Empower through staff walking around the community and talking to potential participants, as well as referral from other participants. The project was explained in detail to participants by trained data collectors, who were not Project Empower staff, and everyone was given time to ask questions about the research before giving their written consent to participate in the survey.”

2. Did the authors consider using log binomial regressions to calculate relative risk ratios instead of odds ratios? I would suggest looking into this since the outcome of experiencing any traumatic event is not rare.	Thank you for raising this. For the revised model we first ran log binomial regression models. The unadjusted logistic regression models and unadjusted log binomial regression produced similar results (statistically significant factors), however there were convergence issues with the adjusted models (most likely due to the presence of continuous covariate and multiple polytomous covariates (Barros & Hirakata 2003)). We therefore modelled the relationships using Poisson models (as they can also produce RRR). Poisson models had no convergence issues, and as such we present that alongside the logistic regressions. We have now added the unadjusted/adjusted RRR to Table 3 in the revised manuscript. We have noted this in the methods section too.
3. In Table 3, the authors should consider using a Bonferroni correction or another correction method to reduce the likelihood of spurious associations due to many analytical models using the same data.	We acknowledge the reviewer's point about likelihood of spurious associations and the suggestion for p-value adjustment. However, we are of the opinion that p-value adjustment is not necessary in our case as i) all the models are using the same outcome (IPV) but different exposures (Armstrong 2014), and ii) that we are looking at the associations here as exploratory analysis for the SEM, where individual trauma items have been summarised into a score.
4. I am wondering if the authors tried excluding "witnessed an armed attack" from the any trauma variables in Table 3. "Witnessed an armed attack" is significant at $p < 0.05$, even though the odds ratio overlaps 1.00. I think it is important to see if that's what is driving the association between "any trauma excluding non-partner rape" with IPV, since the other individual traumatic events are not significantly associated with IPV – it's not intuitive that "any trauma excluding non-partner rape" would be significantly associated with IPV unless essentially solely driven by witnessing an armed attack, in which case, this would have implications for framing the results and these analyses.	Thank you for raising this. In the initial model we felt it important to include all items assessing trauma in one model for the combined variable, because of the potential for significant overlap between items, and because we were interested in the cumulative burden of traumatic events. Based on this suggestion we have done some further analysis to explore whether 'witnessing an armed attack' is driving the association with IPV. While this variable is significant in the unadjusted analysis, so is "witnessing rape" ($p=0.024$), and many of the others are close to 0.05 (i.e. 'robbed' $p=0.063$). In addition, 'witnessing an armed attack' is not significantly associated with IPV in the adjusted analysis ($p=0.117$). Our additional analysis suggests that rather than IPV being driven by one particular type of event, the association with IPV reflects the cumulative burden for women of experiencing different types of traumatic events (with a distinction, as we have

	outlined for non-partner rape).
The same applies to the 3-category variable of number of different traumatic events – could the authors conduct further analyses to determine if the majority of those experiencing 1 or 2 traumatic events are those who experienced non-partner rape and witnessed an armed attack? This may help paint a more accurate picture and allow for a more specific discussion of the results. Relatedly, if “witnessing an armed attack” is driving the association between “any trauma excluding non-partner rape” and IPV, the authors could create a 4-level variable – experienced non-partner rape (and did not witness an armed attack), witnessed an armed attack (and did not experience non-partner rape), both, and neither – to investigate these associations further.	We have done some additional analysis to also investigate this in more detail. Our analysis shows that the women who experienced non-partner rape were not necessarily the same as those who experienced an armed attack (only 13% of our sample experienced both of these events; whereas an additional 21% witnessed an armed attack without experience non-partner rape, and an additional 17% experienced non-partner rape without witnessing an armed attack. As such the numbers are too small to model a 4 level variable.
5. I recommend the authors add to the discussion of poorer mental health as a pathway of traumatic life events to IPV, specifically, how the promotion of and support for better mental health may be linked with self-worth, resilience, empowerment, and more gender equitable views, which in turn could be linked to a reduction in the experiences of IPV. Though I would be careful to phrase this in a way that is not victim blaming, but rather supportive of women and women’s health. This could also be mentioned as an area for future research.	This is a really helpful suggestion and an important point to make – thank you. We have added the following text to our discussion section: “...It also points to a need for further research to investigate positive mental health supports (i.e. engagement coping strategies, gender awareness, etc) that could be provided to women following experiences of non-partner rape that may help in reducing their risk of IPV.
6. The authors may want to incorporate into the discussion (or mention that future research is needed) as to how these results link to the literature on IPV perpetration and male partners. Is it possible that the men who are perpetrating non-partner rape are also perpetrating IPV? This would have implications for interventions to reach these men to stop/reduce their perpetration of violence, whether related to their partners, non-partners, or both.	Thank you for this comment. We agree that the links between IPV perpetration and non-partner rape is important to consider and have now added the following sentence to our conclusion: “Attention to men’s trauma is equally important for IPV reduction interventions, given evidence that the men who commit non-partner rape are often the same as those who perpetuate IPV and that this is often associated with their own experiences of child abuse and neglect (Jewkes et al 2020).”
7. In the first sentence of the conclusions, I suggest adding language to claim that reducing non-partner rate is important in and of itself, rather than focusing on it as a means of reducing IPV. This sentence may come off as a bit insensitive as written. For example, the authors could write: “Our results highlight the need for interventions that comprehensively address multiple forms of violence, with particular attention to non-partner rape, as a means of reducing this traumatic event in and of itself, and as a means of reducing IPV.”	Thank you for the suggestion. We have made this change in the manuscript as suggested.

VERSION 2 – REVIEW

REVIEWER	Watkins, Laura Emory University School of Medicine, Psychiatry and Behavioral Sciences
REVIEW RETURNED	15-Feb-2022

GENERAL COMMENTS	No further comments.
----------------------

REVIEWER	McClair, Tracy Population Council
REVIEW RETURNED	15-Mar-2022

GENERAL COMMENTS	I am pleased with the authors responses and their edits have further strengthened this manuscript. I have no further comments.
--